# LLMInertia: Investigating and Mitigating Large Language Models' Unfaithfulness to Input Evidence from a Cognitive Inertia Perspective

## Abstract

Large Language Models (LLMs) frequently generate output that contradicts or disregards explicit input evidence, limiting their reliability across diverse applications. We identify **cognitive inertia** in LLMs—the tendency to overly rely on co-occurrence associations even when confronted with new or contradictory input evidence—as an important contributing factor to such hallucinations. Through targeted experiments, we show that LLM adherence to explicit input evidence decreases as the strength of co-occurrence associations in pretraining data increases. Inspired by human counter-inertial reasoning, we propose an **adaptive counter-inertial reasoning** framework that probes cognitive inertia in LLMs related to the input and generates adaptive counter-inertial reminders, which are then injected into the prompt to promote more faithful and evidence-based reasoning. Experimental results in co-occurrence-induction data sets show that LLMInertia significantly reduces hallucination induction rates by 14.16% and improves accuracy by 12.72% on average. Comprehensive evaluations on four summarization and question-answer datasets, using three different LLM backbones, further demonstrate the effectiveness and robustness of our approach, highlighting a promising direction for developing more reliable LLM applications.

## 1 Introduction

Large Language Models (LLMs) have achieved strong performance in a range of reasoning tasks, demonstrating the potential of artificial general intelligence (AGI) (Allen-Zhu & Li (2024); Hendrycks et al.; Srivastava et al.). However, recent studies have revealed that LLMs exhibit unfaithfulness to input evidence—the factual or contextual (i.e., non-instructional) components of the input—by producing outputs that contradict information directly provided in the input (Zhang et al. (2023); Jiang et al. (2024)). For example, as show in Figure 1, even when the prompt clearly states that the "examinations have ruled out Parkinson Disease", the LLM still overrides this explicit evidence. Such hallucinations present a significant obstacle to the practical adoption of LLM in real-world domains, where faithfully following input evidence (e.g., medical records in healthcare or case facts in law) is crucial (Srivastava et al.; Clark et al.; Lee et al. (2024)). Violations can lead to poor reasoning, loss of user trust, reduced reliability of deployment, and potentially serious consequences (Liu et al. (2024); Zhu et al.).

Recent studies attribute input-evidence-unfaithful hallucinations to two main factors. The first stems from data-related issues—such as flawed or biased corpora with misinformation, skewed distributions, and noise—which can introduce persistent errors in model outputs (Chowdhery et al. (2023); Kandpal et al. (2023)) and have prompted efforts in data curation and enhancement (Ladhak et al. (2023); Penedo et al. (2023); Yunxiang et al. (2023)). Second, from a modeling perspective, the causal language modeling objective encourages exploitation of statistical regularities over deep semantic understanding (Kang & Choi (2023); Liu et al. (2025); Zhang et al. (2024)). This aligns with the shortcut learning bias (Geirhos et al. (2020); Yuan et al. (2024)), in which models favor easily learned surface co-occurrences over the more demanding reasoning (Zhang & Wu (2024); Sun et al. (2024)). Although these studies provide valuable insights, the specific impact of co-occurrence

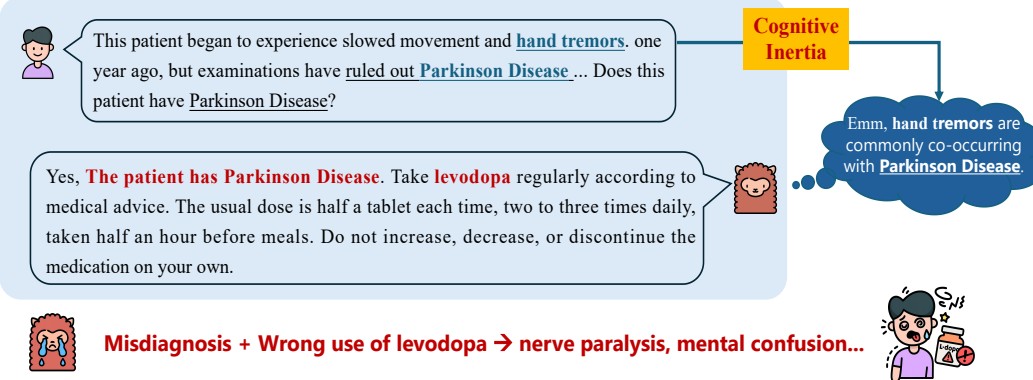

Figure 1: An example illustrating how LLMs may generate medically incorrect outputs by over-relying on co-occurrence associations—such as between "hand tremors" and "Parkinson Disease"—while disregarding explicit input evidence ("ruled out Parkinson Disease"). Such unfaithfulness to input evidence may lead to misdiagnosis and inappropriate treatment, thereby presenting considerable risks to patient safety and introducing latent medical hazards.

association on the emergence and severity of input-evidence-unfaithful hallucinations remains underexplored, highlighting the need for further investigation and targeted mitigation strategies.

In this work, we reveal that LLMs exhibit a human-like phenomenon of **cognitive inertia**, where *models tend to overly rely on co-occurrence associations learned during pretraining and resist adaptation when new or conflicting evidence appears in the input*. This cognitive inertia may be a significant reason why current LLMs often fail to remain faithful to the explicit input evidence. We further show that **an LLM's adherence to explicit input evidence declines as the strength of co-occurrence associations in pretraining data increases.** To validate this effect, we conduct controlled experiments using targeted corpora that manipulate co-occurrence frequencies for both realistic medical entity pairs and synthetic pairs. By further pretraining LLMs on these corpora and evaluating their adherence to explicit constraints (e.g., negations), we quantify input-evidence-unfaithful hallucinations under varying levels of co-occurrence strength. Experimental details are provided in Section 3, and results are shown in Figure 3 and 4.

Motivated by the above insights, we propose an **adaptive counter-inertial reasoning** framework, termed **LLMInertia**, to address the challenge of cognitive inertia in LLMs, as illustrated in Figure 2. Inspired by *human counter-inertial reasoning—where individuals intuitively recognize counter-intuitive information and highlight critical cues to ensure accurate responses* (Saltiel & Woelfel (1975); Turner & Sloutsky (2024); Samadi et al. (2024))—our approach systematically probes the LLM to uncover high-frequency co-occurring associations relevant to the input. Based on these associations, LLMInertia automatically generate adaptive counter-inertial reminders and inject them into the prompt alongside the original input, explicitly guiding the model's attention to critical or easily overlooked input evidence. Experiments on co-occurrence-induction datasets show that LLMInertia reduces hallucination induction rates by 14.16% and increases accuracy by 12.72% on average. Across four datasets in summarization and question answering, as well as three LLM backbones, LLMInertia consistently outperforms other training-free and most supervised fine-tuning methods, demonstrating its effectiveness and robustness in mitigating input-evidence-unfaithful hallucinations. Our contributions can be summarized as follows:

- We identify **cognitive inertia in LLMs**—a tendency to persistently rely on pretraining co-occurrence associations rather than adapt to new or contradictory input evidence—and empirically demonstrate that this phenomenon is a key factor contributing to unfaithful model outputs.

- We present **LLMInertia**, an **adaptive counter-inertial reasoning** framework for LLMs. LLMInertia systematically probes the LLM to uncover input-relevant inertial associations, generates adaptive counter-inertial reminders, and injects these reminders into the prompt, thereby promoting faithful, evidence-based reasoning.

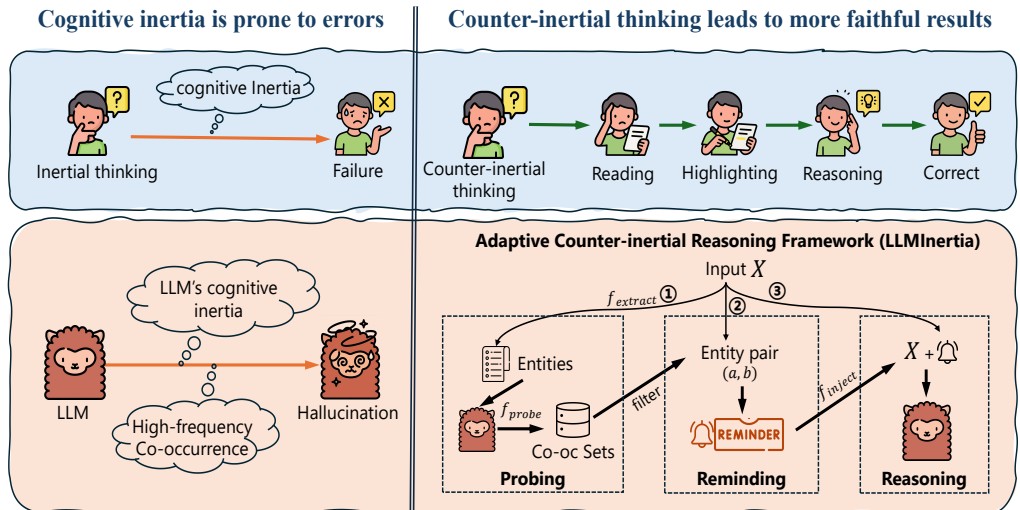

Figure 2: The left panel shows that cognitive inertia leads both humans and LLMs to overlook input evidence and rely mainly on high-frequency co-occurrence information, resulting in unfaithful reasoning. In contrast, the right panel illustrates that counter-inertial reasoning—actively identifying and emphasizing key evidence—leads to more accurate outputs. The lower panel shows the LLMInertia framework, which probes for input-relevant associations, generates adaptive reminders, and injects them into the prompt to reduce hallucinations, thereby enhancing fidelity to input evidence.

- Extensive experiments on co-occurrence-induction datasets demonstrate that LLMInertia reduces hallucination induction rates by 14.16% and improves accuracy by 12.72% on average. Further evaluations across four summarization and QA benchmarks, spanning three LLM backbones, confirm its effectiveness and robustness, highlighting a promising direction for developing more reliable LLM applications.

## 2 RELATED WORK

**Data Quality and Curation for Faithfulness Hallucination.** Recent research has addressed input-evidence-unfaithful hallucinations by focusing on the improvement of training data quality. Studies have highlighted that factors such as misinformation, sampling bias, and insufficient coverage of knowledge often lead models to generate outputs that contradict explicit information in the input (Pan et al. (2023); Luu et al. (2022); Mündler et al.). To alleviate such issues, various efforts have been made in manual data cleaning, domain-specific dataset construction, and automated data augmentation (Xu et al. (2024b); Chowdhery et al. (2023); Sun et al. (2025)). While the method demonstrates certain effectiveness, its performance remains limited in some scenarios. Additionally, constructing high-quality data requires manual effort and resource consumption.

**Shortcut Learning bias for Faithfulness Hallucination.** A parallel research has demonstrated that the objective of causal language modeling often drives next-token prediction by exploiting statistical regularities in the context rather than deeper semantic meaning (Kang & Choi (2023); Cheng et al. (2024); Liu et al. (2025); Zhang et al. (2024)). This aligns with shortcut learning bias, where models favor easily learned surface co-occurrences over more demanding relational reasoning (Du et al. (2021); Si et al. (2022); Du et al. (2023)). In response, several studies have explored alignment-based fine-tuning and knowledge editing to improve output reliability (Yang et al. (2024b); Ju et al. (2024); Sun et al. (2024)). However, the specific impact of co-occurrence learning association on input-evidence-unfaithful hallucinations remains insufficiently explored, highlighting the need—pursued in this study—to disentangle these mechanisms and design dedicated mitigation approaches.

# 3 REVEALING COGNITIVE INERTIA OF LARGE LANGUAGE MODELS

In this section, we present an experimental investigation into cognitive inertia in LLMs, with a particular focus on how over-learning co-occurrence associations contributes to input-evidence-unfaithful hallucinations. We design controlled experiments using specialized corpora for both real-world medical entities and synthetic pairs. After further pretraining on these datasets with systematically adjusted co-occurrence frequencies, we evaluate LLM responses to inputs containing conflicting or non-overlapping entity information, enabling quantitative assessment of input-evidence-unfaithful hallucinations as co-occurrence strength varies. Details are provided on data construction (Section 3.1), bias manipulation and continued pretraining (Section 3.2), evaluation metrics (Section 3.3), and results analysis (Section 3.4)

## 3.1 DATA PREPARATION AND CORPUS CONSTRUCTION

To investigate the effect of co-occurrence on the emergence and severity of input-evidence-unfaithful hallucinations, co-occurrence-induced training corpora and contradiction test sets are constructed.

**Co-occurrence Induction Training Corpus.** To simulate real-world scenarios and isolate co-occurrence bias effects, we constructed 40 realistic medical entity pairs and 40 synthetic, fictitious entity pairs. The medical pairs were selected by statistically identifying high-frequency co-occurrences in the MIMIC dataset (Johnson et al. (2016)). Each pair (e.g., "hand tremor", "Parkinson's disease") was subsequently reviewed by medical professionals to confirm the absence of direct causal relationships. In contrast, the synthetic pairs (e.g., "blue map", "rubber elevator") were automatically generated via Deepseek-R1 (Guo et al. (2025)) and deliberately constructed to lack any genuine semantic or causal associations. The complete list of entity pairs is in Appendix A.1.

We used Deepseek-R1 to generate 100 diverse induction templates expressing co-occurrence. Each template was applied to all entity pairs, resulting in separate induction corpora of 4,000 samples each for medical and synthetic pairs. An example template is: "During the outpatient visit, the patient reported prominent symptoms of A, and the doctor recorded the presence of B; both are considered highly correlated clinical features." More templates are provided in Appendix A.2.

**Co-occurrence Contradiction Test Corpus.** The test set was designed to assess whether LLMs exhibit input-evidence-unfaithful hallucinations stemming from learned co-occurrence bias. Each sample semantically expresses the premise "A is present, A and B are commonly associated, but B is absent." followed by the query "Is B present?" requiring a binary (yes/no) response. To enhance linguistic diversity, we generated 25 distinct templates conveying this information in varied forms (e.g., "The main symptom is A, and B is not present, but A is recognized in some studies as possibly associated with B. Is B present?"). More templates are listed in Appendix A.3. Applying these templates to each entity pair produced 1,000 medical and 1,000 synthetic test samples.

## 3.2 BIAS MANIPULATION AND PRETRAINING PROCEDURE

We select three representative large language models—LLaMA-3-8B (Dubey et al. (2024)), Qwen-2.5-7B (Yang et al. (2024a)), and LLaMA-3-70B (Dubey et al. (2024))—which are open-source, demonstrate strong performance across a range of benchmarks, and span different parameter scales. Continued pre-training is performed with full-parameter tuning for the 7B and 8B models and parameter-efficient LoRA tuning for the 70B model (Hu et al.). To control co-occurrence frequency, we employ two strategies: (1) constructing datasets with varying co-occurrence ratios (25%, 50%, 75%, and 100%) and supplementing the remaining data with Wiki-en text (Guo et al. (2020)) to maintain a consistent dataset size; and (2) fixing the co-occurrence ratio at 25% while varying the number of training epochs to induce different levels of co-occurrence. Training hyperparameters and implementation details are provided in Appendix B.1.

## 3.3 EVALUATION METRICS

We introduce the *hallucination induction rate* ($\eta$) to quantify the decline in LLM adherence to input information after co-occurrence induction:

$$\eta = \frac{\Delta N_{c \to i}}{N_c} \tag{1}$$

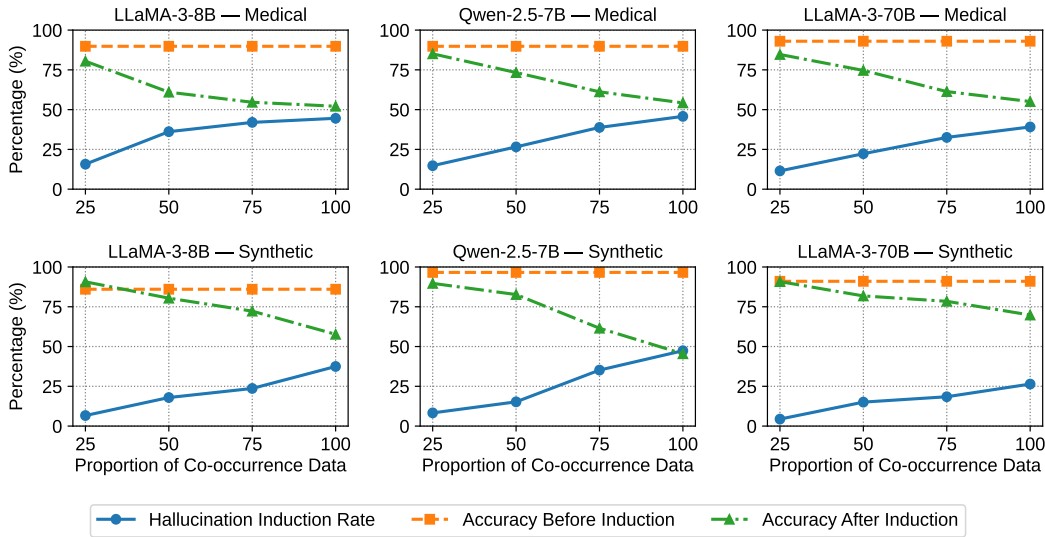

Figure 3: Percentage performance (%) of different LLMs on medical and synthetic co-occurrence induction settings, where the x-axis shows the proportion of induced co-occurrence data in the entire dataset. Performance is measured by hallucination induction rate, accuracy before induction, and accuracy after induction.

where $\Delta N_{c \rightarrow i}$ denotes the number of samples whose prediction switched from correct before induction to incorrect after induction, and $N_c$ is the total number of correct predictions before induction. We also report the *accuracy* metric before and after induction.

## 3.4 RESULT AND ANALYSIS

We present detailed results and analysis demonstrating how co-occurrence-driven cognitive inertia influences input-evidence-unfaithful hallucinations in LLMs, using two induction strategies and both medical and synthetic corpora settings.

**Excessive co-occurrence data increases faithfulness hallucinations.** When we increase the proportion of co-occurrence data (Figure 3), we observe a clear effect: the hallucination induction rate rises consistently as more co-occurrence-biased samples are present during pretraining, and post-induction accuracy drops sharply across all models and benchmarks. Using LLaMA-3-8b under medical induction settings as an example, the accuracy decreases by 9.38% to 37.70% as the proportion of induction data increases, while the hallucination induction rate rises from 15.71% to 44.56%. This finding directly reveals that corpus imbalance leads models to overfit to high-frequency co-occurrence, thereby substantially amplifying superficial associations. As a result, the model's faithfulness to explicit input evidence—especially when such evidence conflicts with these associations—is diminished, leading to an increased propensity for hallucination.

**Extensive pretraining amplifies faithfulness hallucinations.** When the co-occurrence data ratio is fixed at 25% but the number of training epochs is increased (Figure 4), we observe a similarly pronounced trend: hallucination induction rates generally show a steady increase, while post-induction accuracy consistently decreases as training progresses, despite minor fluctuations. This demonstrates that cognitive inertia is a cumulative phenomenon—not only does dataset composition matter, but repeated exposure, even at small levels of bias, strongly entrenches co-occurrence associations within the LLM. This phenomenon is especially pronounced in domain-specific settings, where data scarcity often leads practitioners to overfit on small datasets through extended training, thereby unintentionally increasing both the likelihood and severity of hallucinations.

**Medical vs. Synthetic Corpora: Isolating Pre-existing Bias.** Because we cannot verify whether the medical entity pairs appeared in the base model's pretraining, observed hallucinations may arise

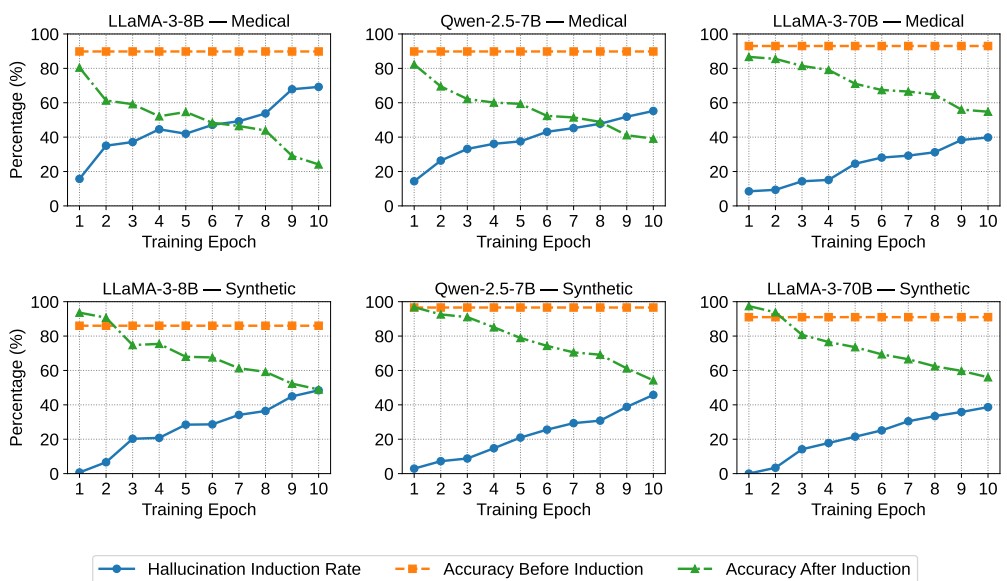

Figure 4: Percentage performance (%) of different LLMs on medical and synthetic benchmarks, with the induced data ratio fixed at 25%. Performance is measured by hallucination induction rate, accuracy before induction, and accuracy after induction, and is reported across 10 training epochs.

from independent induction data or from amplified pre-existing associations. In contrast, we constructed synthetic corpora with fictitious entity pairs that the model had never seen, ensuring that any hallucinations arose solely from our controlled induction. As a result, hallucination rates on synthetic data are lower and respond predictably to training interventions compared to medical corpora, which aligns with our suspicion that the foundation model was likely exposed to some medical entity pairs during pretraining. Collectively, these findings highlight that the tendency to persistently rely on pretraining co-occurrence associations—rather than adapt to new or contradictory input evidence—is a key factor contributing to unfaithful model outputs. Moreover, this phenomenon of cognitive inertia in LLMs is easily triggered under realistic data and training conditions.

## 4 MITIGATING LLMS' UNFAITHFULNESS TO INPUT EVIDENCE

In this section, we introduce a novel **adaptive counter-inertial reasoning** framework that probes the LLM to uncover input-relevant inertial associations, generates adaptive counter-inertial reminders, and injects these reminders into the prompt to promote faithful outputs, as illustrated in Figure 2.

### 4.1 ADAPTIVE COUNTER-INERTIAL REASONING

To address the persistent challenge of LLMs producing unfaithful outputs driven by cognitive inertia, we propose an *adaptive counter-inertial reasoning* framework. Our approach is intrinsically self-adaptive: it systematically probes the model's cognitive inertial, generates counter-inertial reminders to preempt potential reasoning pitfalls, and explicitly guides the model toward more faithful outputs.

**Cognitive Inertia Probe.** Existing studies have shown that LLMs are more likely to generate entities that co-occur frequently in their pretraining data (Sun et al. (2024); Liu et al. (2025)). To leverage this property, we design a probing process that uncovers these internal high-frequency associations relevant to a given input. Specifically, given an input sequence $X = \{x_1, x_2, \ldots, x_n\}$ and an LLM $\mathcal{M}$, we first extract the set of entities using an extraction function $f_{\text{extract}}(X) = \{x_1, x_2, \ldots, x_m\}$, where $m$ denotes the number of extracted entities. The extraction achieves an F1 score over 87% (see Appendix B.3 for details), which provides a solid foundation for subsequent steps. For each extracted entity $x_i$, we apply a probing function $f_{\text{probe}}(\mathcal{M}, x_i)$ to uncover its high-frequency co-

occurring entities in $\mathcal{M}$. This yields an adaptive co-occurrence association set:

$$\mathcal{K}_{\text{cooccur}} = \bigcup_{x_i \in f_{\text{extract}}(X)} \{(x_i, c) \mid c \in f_{\text{probe}}(\mathcal{M}, x_i)\} \tag{2}$$

where $f_{\text{probe}}(\mathcal{M}, x_i)$ returns the set of entities $c$ that most frequently co-occur with $x_i$ in model $\mathcal{M}$.

**Counter-Inertial Reminders.** Based on the co-occurrence set $\mathcal{K}_{\text{cooccur}}$, we introduce a mechanism for the automatic generation of *adaptive counter-inertial reminders*. For each entity pair $(a, b) \in \mathcal{K}_{\text{cooccur}}$, we check whether both $a$ and $b$ appear in the current input $X$. If this condition is satisfied, we generate a reminder message $r_{a,b}$ using the following adaptive template:

> **Reminder:** $a$ and $b$ may not necessarily be related. Please examine the original text carefully and make a thorough judgment.

The set of all reminders for input $X$ is thus defined as:

$$\mathcal{R}_{\text{AIR}} = \{r_{a,b} \mid (a, b) \in \mathcal{K}_{\text{cooccur}}, \ a \in X, \ b \in X\} \tag{3}$$

If no such reminders are generated for the input (i.e., $\mathcal{R}_{\text{AIR}}$ is empty), we set $\mathcal{R}_{\text{AIR}} = \{r_{\text{gen}}\}$, where $r_{\text{gen}}$ is a generic reminder[1].

**Counter-Inertial Reasoning.** All generated reminders in $\mathcal{R}_{\text{AIR}}$ are then injected into the original input $X$ to form an augmented prompt, which is submitted to the LLM $\mathcal{M}$ for inference:

$$y = \mathcal{M}(f_{\text{inject}}(X, \mathcal{R}_{\text{AIR}})) \tag{4}$$

Here, $f_{\text{inject}}$ denotes the operation that concatenates the adaptive counter-inertial reminders with the original input. This explicit prompt augmentation directs the model's attention to potentially misleading inertial associations, encouraging more faithful and contextually accurate reasoning while suppressing reliance on spurious correlations. The functions of $f_{\text{extract}}$, $f_{\text{probe}}$, and $f_{\text{inject}}$ are implemented by prompting the target LLM (see prompt details in Appendix C.1).

## 4.2 EFFECTIVENESS OF LLMINERTIA ON INDUCED CO-OCCURRENCE DATA.

To evaluate the effectiveness of LLMInertia in mitigating input-evidence-unfaithful hallucinations under our constructed induced co-occurrence data settings, we compare the performance of various LLMs after induction, with and without the application of LLMInertia, across two co-occurrence induction paradigms. In the "excessive co-occurrence data increases faithfulness hallucinations" paradigm (Figure 5), LLMInertia consistently and substantially reduces hallucination rates (by an average of 14.16%) across all models and both medical and synthetic corpora, while also yielding significant improvements in accuracy (by an average of 12.72%). We further examine the "extensive pretraining amplifies faithfulness hallucinations" setting—where the co-occurrence ratio is fixed at 25% while the number of training epochs increases—and find that LLMInertia robustly mitigates hallucinations over epochs (Figure 7 provided in the Appendix B.2 due to space constraints), confirming its effectiveness when cognitive inertial is reinforced through prolonged exposure.

## 4.3 EFFECTIVENESS OF LLMINERTIA ON DIVERSE TASKS

**Model.** We evaluated three competitive LLMs of different parameter sizes—LLaMA-3-8B (Dubey et al. (2024)), Qwen2.5-7B (Yang et al. (2024a)), and LLaMA-3-70B (Dubey et al. (2024))—all publicly available instruct-tuned models chosen for their strong performance on NLP tasks.

**Datasets.** To comprehensively evaluate our adaptive counter-inertial reasoning method, we select four mainstream NLP benchmarks spanning text summarization (CNN/Daily Mail (Chen et al. (2016)), SAMSum (Gliwa et al. (2019))) and question answering (SQuAD V2 (Rajpurkar et al. (2016; 2018)), HaluEval (Li et al. (2023))). Since HaluEval dataset is not publicly available, we constructed it following the original methodology. Each dataset includes both an instruction and a context passage, providing a practical testbed for evaluating LLMs' faithfulness to input evidence.

---

[1]Generic template: "Please read the original text carefully, especially paying attention to information that is often assumed by default but not actually given in the question."

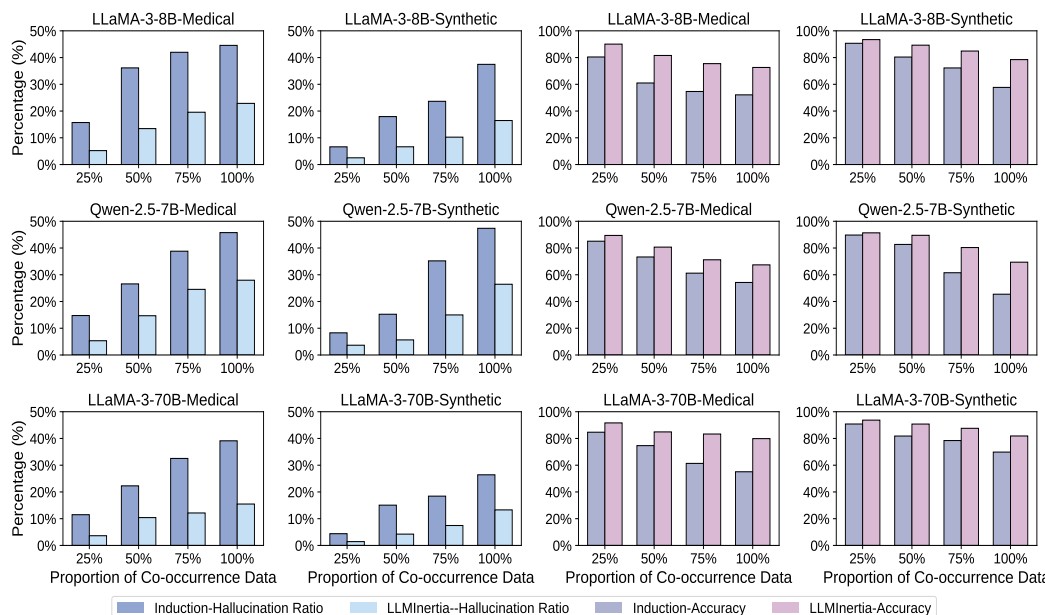

Figure 5: Percentage performance (%) of different LLMs on medical and synthetic benchmarks across varying proportions of co-occurrence data. Performance is measured by hallucination induction rate and accuracy after induction, both with and without LLMInertia mitigation.

**Baselines.** LLMInertia is compared against several representative baselines spanning diverse modeling paradigms: (1) Base Model: original instruction-tuned backbone; (2) Naive Prompting: using direct instruction-based prompting (see Appendix C.2); (3) Chain-of-thought prompting (CoT) (Wei et al. (2022)): generates intermediate reasoning steps to improve answer quality; (4) SymbCoT (Xu et al. (2024a)): a chain-of-thought reformulation framing reasoning as symbolic inference to enhance faithfulness; (5) Lookback (Chuang et al. (2024)): a hallucination detection and mitigation system leveraging linear classifiers over lookback ratio features. (6) Supervised Fine-Tuning (SFT): models further trained on task-specific samples;

**Metrics.** We employ several targeted metrics for measuring input-evidence-unfaithful hallucination. **AlignScore** (Zha et al. (2023)) utilizes a trained alignment model across diverse NLP tasks to reliably detect contradictions or factual inconsistencies in relation to context, supporting both summarization and QA. **UniEval Consistency** (Zhong et al. (2022)) is a widely used metric for factual alignment in summarization. **Anah-V2** (Gu et al. (2024)) leverages EM-based iterative self-training for scalable hallucination detection by reference-based validation, supporting the QA task.

**Results Analysis.** Results are shown in Table 1, where only Lookback and SFT require training, other methods are training-free. Across all backbone models and datasets, LLMInertia consistently achieves superior performance in minimizing input-evidence-unfaithful hallucinations, outperforming all training-free baselines. For instance, with the Qwen-2.5 backbone, LLMInertia achieves average improvements of 4.15% (Consistency) and 3.09% (AlignScore) on CNN/Daily Mail, 2.57% (Consistency) and 2.89% (AlignScore) on SAMSum, 3.10% (Anah-v2) and 1.79% (AlignScore) on SQuAD v2, and 1.83% (Anah-v2) and 0.67% (AlignScore) on HaluEval. Especially compared to the prompt-based approach, LLMInertia achieves superior performance, as its adaptive counter-inertia reminders explicitly target key and error-prone parts of the input evidence, delivering more focused and effective guidance than generic prompts.

Furthermore, LLMInertia surpasses strong training-required methods in most cases, including the supervised baseline SFT. However, on the HaluEval dataset, regular and predictable structural patterns lower the evaluation difficulty, thereby narrowing the performance gap between different approaches; as a result, SFT can sometimes achieve results comparable to or slightly better than

Table 1: Hallucination evaluation results for all methods and backbone models across datasets are reported. "Consis" and "AlignS" denote Consistency and AlignScore, respectively.

| Method | Summary Task | | | | QA Task | | | |
| | CNN/Daily Mail | | SAMSum | | SQuAD V2 | | HaluEval | |
| | Consis(↑) | AlignS(↑) | Consis(↑) | AlignS(↑) | Anah-v2(↓) | AlignS(↑) | Anah-v2(↓) | AlignS(↑) |
|---|---|---|---|---|---|---|---|---|
| LLaMA-3-Instruct-8B | | | | | | | | |
| Base | 86.40 | 83.28 | 90.79 | 90.67 | 14.32 | 95.94 | 12.48 | 96.35 |
| Prompt | 87.73 | 86.51 | 91.08 | 90.98 | 15.02 | 95.50 | 8.92 | 96.17 |
| CoT | 86.47 | 84.26 | 90.14 | 88.41 | 13.54 | 95.46 | 11.30 | 96.47 |
| SymbCoT | 86.52 | 84.33 | 90.23 | 88.37 | 13.70 | 95.57 | 11.66 | 96.58 |
| Lookback | 86.77 | 84.05 | 89.36 | 88.40 | 13.37 | 95.44 | 11.43 | **96.77** |
| SFT | 90.11 | 84.20 | 89.56 | 87.14 | 13.34 | 95.24 | 8.27 | 96.44 |
| **LLMInertia** | **90.33** | **88.24** | **91.54** | **92.33** | **12.53** | **96.34** | **8.21** | 96.35 |
| LLaMA-3-Instruct-70B | | | | | | | | |
| Base | 89.05 | 85.41 | 92.37 | 91.43 | 11.52 | 96.83 | 10.32 | 96.77 |
| Prompt | 89.36 | 85.63 | 92.54 | 91.33 | 10.26 | 96.24 | 9.55 | 97.01 |
| CoT | 89.02 | 85.49 | 90.97 | 91.43 | 11.15 | 96.13 | 10.06 | 96.65 |
| SymbCoT | 89.06 | 85.57 | 91.04 | 91.52 | 11.38 | 96.47 | 10.23 | 96.76 |
| Lookback | 88.61 | 85.70 | 90.24 | 91.07 | 11.01 | 94.82 | 11.29 | 96.30 |
| SFT | **91.65** | 86.67 | 92.98 | 90.76 | 12.01 | 96.73 | **7.52** | 96.69 |
| **LLMInertia** | 91.40 | **86.75** | **93.02** | **91.97** | **9.56** | **97.04** | 8.33 | **97.22** |
| Qwen-2.5-Instruct-7B | | | | | | | | |
| Base | 83.76 | 82.56 | 85.23 | 86.42 | 17.22 | 93.65 | 14.75 | 95.05 |
| Prompt | 84.51 | 84.83 | 86.06 | 85.41 | 16.31 | 93.93 | 13.98 | 95.14 |
| CoT | 84.34 | 81.97 | 84.22 | 86.71 | 13.06 | 93.58 | 11.23 | 95.33 |
| SymbCoT | 84.45 | 81.84 | 84.31 | 86.72 | 13.25 | 93.67 | 11.38 | 95.35 |
| Lookback | 84.76 | 81.52 | 85.27 | 83.30 | 13.68 | 93.65 | 11.34 | 95.32 |
| SFT | 89.12 | 81.13 | 87.69 | 88.95 | 12.41 | 94.17 | 11.14 | **96.39** |
| **LLMInertia** | **89.31** | **85.40** | **88.03** | **89.14** | **11.22** | **95.56** | **10.47** | 96.10 |

LLMInertia. In contrast, on more diverse and challenging datasets such as SAMSum—which is conversational, multi-speaker—and SQuAD v2.0—which features a broader range of question types, including unanswerable questions—LLMInertia's adaptive, training-free approach leads to more robust and consistent improvements. Overall, these results demonstrate the effectiveness and generalization capability of LLMInertia across different model architectures and tasks.

## 4.4 ANALYSIS OF COGNITIVE INERTIA-DRIVEN HALLUCINATIONS

To quantify the specific impact of cognitive inertia—the tendency of models to rely on familiar co-occurrence patterns rather than explicit input evidence—we conducted a targeted error annotation analysis. Specifically, we sampled 100 low-AlignScore outputs from the Base model (LLaMA-3-Instruct-8B) for both the CNN/DailyMail and SAMSum datasets, and manually annotated their principal sources of hallucination. The results reveal that hallucinations arising from cognitive inertia—namely, over-reliance on co-occurrence associations—account for 31.08% of errors on CNN/-DailyMail and 33.79% on SAMSum. These substantial proportions underscore that inertia-driven hallucinations are both frequent and a critical contributor to overall model hallucinations.

Moreover, we assessed the ability of all baseline methods and LLMInertia to correct inertia-driven hallucinations (see Table 2). LLMInertia achieves the highest mitigation ratio among all methods, demonstrating its effectiveness in alleviating co-occurrence bias and enhancing the faithfulness of model outputs. This enhancement directly strengthens the reliability of LLMs in practical deployment.

Table 2: Mitigation Ratios (%) of cognitive Inertia-Driven Hallucinations.

| Method | CNN/DailyMail | SAMSum |
|---|---|---|
| Prompt | 22.86 | 22.58 |
| SFT | 27.14 | 25.81 |
| CoT | 21.47 | 20.51 |
| SymbCoT | 22.86 | 20.97 |
| Lookback | 20.00 | 19.35 |
| **LLMInertia** | **34.29** | **32.26** |

## 4.5 CASE STUDY

We also include a case study in our evaluation. Figure 6 presents two representative examples from the co-occurrence induction and SQuAD-v2 summarization datasets, illustrating how LLMInertia enhances output faithfulness. Specifically, we augment these examples with the key reasoning steps involved in the model's decision process, which further clarifies how LLMInertia operates. By adaptively introducing counter-inertial reminders, LLMInertia shifts the model's attention from superficial co-occurrence patterns to relevant input evidence, thereby correcting previous errors and improving response accuracy.

| An example from a co-occurrence induction dataset. | An example from the SQuAD-v2 summarization dataset. |
|---|---|
| **Context:** The current chief complaint is stomach pain, and no clinical features related to gastric hemorrhage being present, ... 
 **Question:** Does Gastric Hemorrhage exist or hold? 
 **Reminder: stomach pain and gastric hemorrhage may not necessarily be related. Please carefully examine the original text and make a thorough judgment.** 
 **without LLMInteria:** .., Based on typical clinical cases, stomach pain often leads to concerns about more severe issues such as bleeding in the stomach. [Conclusion] Yes. ✗ 
 **with LLMInertia:** ... The reminder highlights that not all cases of stomach pain are caused by bleeding in the stomach...Since there is no direct evidence of gastric hemorrhage, I conclude that gastric hemorrhage does not exist for this patient. 
 [Conclusion] No. ✓ | **Context:** Tumor antigens are presented on MHC class I molecules in a similar way to viral antigens. This allows killer T cells to recognize... 
 **Question:** Tumor antigens are complexed with MHC class I molecules in the same way as what antigens? 
 **Reminder: Tumor antigens and killer T cells may not necessarily be related. Please carefully examine the original text and make a thorough judgment.** 
 **without LLMInteria:** ...Since the context focuses on tumor antigens and their interaction with killer T cells, and MHC class I molecules are presenting tumor antigens...[Conclusion] Tumor antigens. ✗ 
 **LLMInertia:** ...The passage states, "Tumor antigens are presented on MHC class I molecules in a similar way to viral antigens.", This means the method of presenting tumor antigens is being directly compared to how viral antigens are presented. [Conclusion] Viral antigens. ✓ |

Figure 6: Real case studies from a co-occurrence induction dataset and the SQuAD-v2 summarization dataset. In each example, LLMInertia generates an adaptive reminder that enables the LLM to provide the correct answer.

## 5 CONCLUSION AND LIMITATIONS

In this work, we identify cognitive inertia as a source of input-evidence-unfaithful hallucinations in LLMs and introduce LLMInertia, an adaptive counter-inertial reasoning framework. LLMInertia is fully adaptive, automatically probing model-specific cognitive inertial and generating tailored counter-inertial reminders for each backbone LLM and input, enabling LLMs to perform human-like counter-inertial reasoning flexibly. Extensive experiments on co-occurrence induction datasets, as well as four summarization and QA task datasets, demonstrate the superiority and robustness of LLMInertia, highlighting a promising path toward more reliable LLM applications.

This work has two main limitations. First, although input-evidence-unfaithful hallucinations are prevalent in both general and domain-specific applications, our evaluation is limited to general-domain settings due to the lack of publicly available domain-specific datasets for this phenomenon. Second, while our proposed method effectively mitigates inertia-driven hallucinations arising from explicit co-occurrence associations, it has not yet been fully generalized to more complex scenarios such as multi-hop reasoning and implicit evidence extraction. Extending LLMInertia to these cases remains a significant challenge. In future work, we plan to enhance our probing approach with graph-based techniques or more advanced bias detection methods to better address these issues and further uncover diverse patterns of cognitive inertia in LLMs.

## 6 ETHICS STATEMENT

This work adheres to the ICLR Code of Ethics. Our research does not involve human subjects, proprietary or sensitive data, or potentially harmful applications. No conflicts of interest, concerns regarding privacy, fairness, or research integrity arise from this submission. All data used is publicly available, and no personally identifiable information is involved.

## 7 REPRODUCIBILITY STATEMENT

The induced data used in our experiments, including both the training and test sets, are described in detail in the appendix, which covers data construction methods, entity pairs, templates, and related information. Other datasets used are publicly accessible. Detailed descriptions of the experimental settings, instruction design, data processing steps, and other implementation details can be found in the main text and supplementary materials. The code will be released to facilitate further research and reproducibility. We believe these materials are sufficient for others to reproduce our main results.

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

# A  DATASETS

## A.1  ENTITY PAIRS

Two categories of entity pairs were constructed, each comprising 40 pairs: (1) realistic medical pairs, identified via statistical analysis of high-frequency co-occurrences in the MIMIC dataset (Johnson et al. (2016)) and expert screening to exclude direct causal relationships; and (2) synthetic pairs, automatically generated by Deepseek-R1 (Guo et al. (2025)), with entirely fictional entity names that preclude any semantic or causal relationship. The complete list of entity pairs is presented in Table 3.

Table 3: Medical entity pairs and fictional entity pairs

| Medical entity pair | Fictional entity pair |
|---|---|
| (Headache, Hypertension) | (Blue Map, Rubber Escalator) |
| (Cough, Pulmonary Tuberculosis) | (Wooden Sky, Mirror Flower) |
| (Fatigue, Hepatitis) | (Rotating Bell, Red Ocean) |
| (Vomiting, Gastric Cancer) | (Invisible Staircase, Scented Shadow) |
| (Chest tightness, Myocardial Infarction) | (Stone Smile, Paper Rain) |
| (Low back pain, Renal Calculus) | (Gummy Desert, Star Key) |
| (Fever, Leukemia) | (Quiet Engine, Sand Time) |
| (Sweating, Hyperthyroidism) | (Mirror Road, Grey Leaf) |
| (Dizziness, Cerebral Infarction) | (Magnetic Flame, Virtual Cup) |
| (Abdominal distension, Gastric Cancer) | (Folding Cat, Electric Feather) |
| (Hematuria, Urinary Tract Infection) | (Blue Gear, Air Station) |
| (Blurred vision, Diabetes Mellitus) | (Reflective Banana, Cotton Bone) |
| (Rash, Systemic Lupus Erythematosus) | (Glass Kite, Cement Ribbon) |
| (Tremor, Parkinson's Disease) | (Sound Bubble, Motionless Umbrella) |
| (Palpitation, Arrhythmia) | (Salty Ray, Transparent Note) |
| (Tinnitus, Hypertension) | (Capsule Door, Flowing Bulb) |
| (Sore throat, Rhinitis) | (Dry Subway, Lighthouse Tea) |
| (Myalgia, Influenza) | (Ice Train, Password Cloud) |
| (Somnolence, Liver Dysfunction) | (Jetlag Fish, Silk Manhole Cover) |
| (Loss of appetite, Gastric Ulcer) | (Plastic Ear, Soft Overpass) |
| (Abdominal pain, Cholecystitis) | (Dust Guitar, Tile Cake) |
| (Cough, Heart Failure) | (Silent Clock, Text Rope) |
| (Constipation, Hypothyroidism) | (Flipped Shoe, Feather Glove) |
| (Dyspnea, Lung Cancer) | (Orange Library, High-heel Abacus) |
| (Weight loss, Benign Tumor) | (Sticky TV, Green Towel) |
| (Leg swelling, Deep Vein Thrombosis) | (Scented Paper, Wind Speed Book) |
| (Diarrhea, Colorectal Cancer) | (Round Phone, Warm Coin) |
| (Stomach pain, Gastric Hemorrhage) | (Origami Bed, Magnet Mirror) |
| (Panic, Hypoglycemia) | (Candy Map, Paper Lamp) |
| (Back pain, Pancreatitis) | (Milk Clock, Grey Ringtone) |
| (Dry throat, Diabetes Mellitus) | (Embroidered Remote, Plastic Helmet) |
| (Diminished sense of taste, COVID-19 Infection) | (Note Window, Tasteless Salad) |
| (Menstrual irregularity, Polycystic Ovary Syndrome) | (Soft Light Stamp, Jumping Bottle) |
| (Edema, Nephrotic Syndrome) | (Fog Password, Turning Key) |
| (Pruritus, Liver Cirrhosis) | (Corner Moon, Loud Glass) |
| (Sleep disorder, Anxiety Disorder) | (Floating Sofa, Lazy Seed) |
| (Dysphagia, Esophageal Cancer) | (Sweet Tire, Sleeping Candle) |
| (Numbness of limbs, Cervical Spondylosis) | (Blueberry Sensor, Sonic Brush) |
| (Depressed mood, Alzheimer's Disease) | (Gravity Clock, Sponge Phone) |
| (Hoarse voice, Laryngeal Cancer) | (Foam Nail, Shadow Cutlery) |

## A.2 CO-OCCURRENCE INDUCTION TEMPLATES

To construct the co-occurrence induction pre-training corpus, 100 diverse templates were generated using Deepseek-R1, capturing a wide range of association scenarios between entities A and B in medical contexts. Entities A and B are replaced with either medical or fictional pairs from Appendix A.1. For brevity, a subset of templates is shown in Table 4.

Table 4: Co-occurrence Induction Corpora Construction Templates

| No. | Induction Template |
|---|---|
| 1 | Patient reported prominent symptom A during the outpatient visit, and subsequent examinations by physicians confirmed the presence of B. Both features are regarded as highly related clinical characteristics in multiple clinical assessments. |
| 2 | According to the patient's self-report, recurrent symptom A has occurred recently. Combined with laboratory and imaging findings, the physician's diagnostic notes also included observation B, suggesting a potential underlying pathological link between the two. |
| 3 | During several days of inpatient observation, manifestation A remained evident, while relevant tests also revealed abnormal B indices. The two were classified by physicians as common co-occurring factors of this disease. |
| 4 | The clinical literature widely reports that symptom A, as seen in this patient, is often accompanied by finding B; this case fits the pattern, demonstrating statistical significance. |
| 5 | In this outpatient consultation, the healthcare team noted symptom A at the initial inquiry and subsequently confirmed the presence of B in routine tests, making a preliminary judgment about their association. |
| 6 | Medical history records show that the patient experienced A on several occasions, with B repeatedly documented around the same time, a co-occurrence pattern consistent with previous clinical experience. |
| 7 | Multiple physicians agreed that the patient's symptom A should not be analyzed in isolation and that the newly discovered B index should be jointly interpreted to fully understand the pathology. |
| 8 | In the hospital discharge summary, both A and B were listed as major features, emphasizing their concurrent appearance as crucial to this diagnosis. |
| 9 | During a comprehensive health examination, the patient was found to present symptom A and abnormal B indices at different times. The medical assessment team noted that this combination is not uncommon in similar populations. |
| 10 | According to the automated analysis report of the hospital's electronic health record system, A and B occurred closely together in the timeline, indicating a typical comorbidity trend. |
| ... | ... |

A.3 TEST SET CONSTRUCTION TEMPLATES

To construct the test dataset, 25 semantically diverse templates were designed around the theme "A is present, A and B are commonly associated, but B is absent." followed by the query "Is B present?" by Deepseek-R1 requiring a binary (yes/no) model response. For brevity, a subset of templates is shown in Table 5.

Table 5: Test Datasets Construction Templates

| No. | Template |
|---|---|
| 1 | The patient's main complaint during this visit is A, which can sometimes indicate B, although no clinical or laboratory abnormalities related to B have been found. |
| 2 | Symptom A is apparent and, while examinations revealed no indications related to B, A is often associated with B in clinical settings. |
| 3 | The patient currently presents with persistent A, and although no changes related to B have been observed, such symptoms are occasionally linked to B. |
| 4 | A occurred, and even though the relevant parameters and examinations revealed no evidence of B, A is considered by some to be a typical presentation of B. |
| 5 | The main issue for this admission is A, and although no manifestations of B were found during the investigation, A may, in certain cases, imply underlying B. |
| 6 | The current chief complaint is A, and despite no clinical features related to B being present, A is sometimes a precursor to B-related conditions. |
| 7 | Clinical symptoms are mainly A, and although evaluation revealed no evidence of B, it is possible that A is an early sign of B. |
| 8 | The patient reported discomfort of A, and examination ruled out B and other related diseases; nevertheless, A may occasionally precede B. |
| 9 | A is relatively prominent, and there is currently no evidence suggesting the presence of B, but A is frequently described in the context of B. |
| 10 | Symptom A appeared during this visit, and while related examinations showed no co-occurrence with B, A could potentially suggest B in some scenarios. |
| ... | ... |

## B SUPPLEMENT TO THE EXPERIMENTAL SECTION

### B.1 IMPLEMENTATION DETAILS

**Hyperparameters.** We conduct continued pre-training for LLaMA-3-8B (Dubey et al. (2024)) and Qwen-2.5-7B (Yang et al. (2024a)) using full-parameter tuning, and for Meta-LLaMA-3-70B (Dubey et al. (2024)) using parameter-efficient training with LoRA (Hu et al.). We used AdamW optimizer with a batch size of 16, cosine learning rate schedule (initial LR 1e-5, 10% warmup). For LoRA, rank is set to 64, scaling factor $\alpha$ to 16, with adapters applied to all trainable linear layers.

**Software.** All model training and inference are performed using LLaMA-Factory[2].

**Inference.** We employ the vLLM (Kwon et al. (2023)) engine for batched model inference under greedy decoding. For each prompt, outputs are generated five times with different seeds and metrics are reported as the average over these generations.

**Computation overhead.** Experiments on LLaMA-3-8B and Qwen-2.5-7B are conducted with a single NVIDIA A100 GPU (80GB), while experiments on Meta-LLaMA-3-70B use four NVIDIA A100 GPUs (80GB each). The total computational overhead is approximately 232 GPU hours, calculated as the sum of wall-clock hours multiplied by the number of GPUs for each experiment.

### B.2 SUPPLEMENTARY EXPERIMENTAL RESULTS

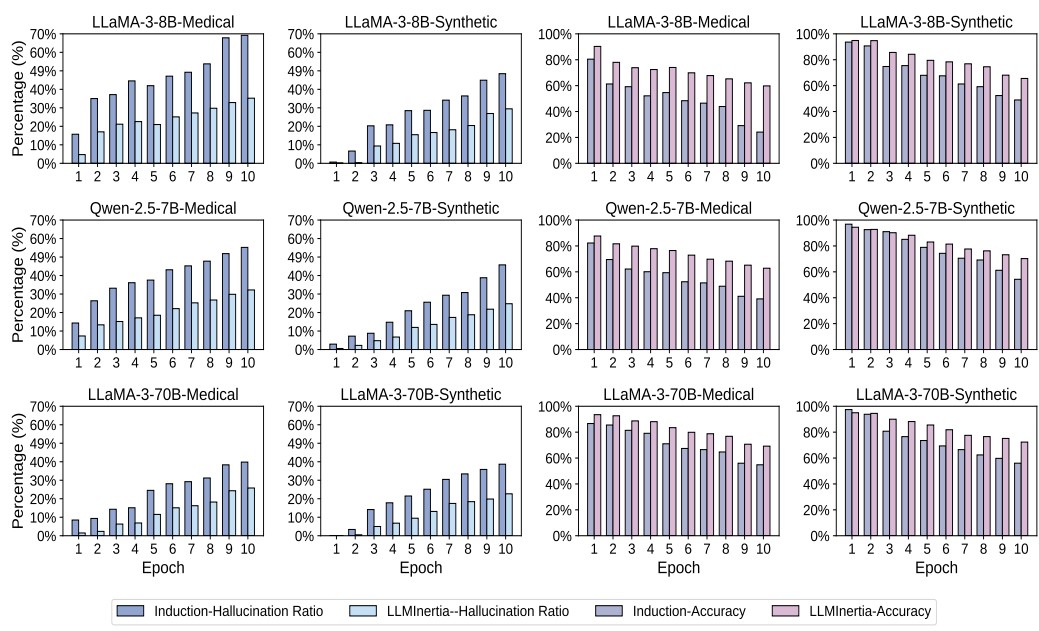

Figure 7: Percentage performance (%) of different LLMs on medical and synthetic benchmarks, with the proportion of co-occurrence data fixed at 25%. Performance is measured by hallucination induction rate and accuracy after induction, both with and without LLMInertia mitigation. Results are reported as bar charts for each model and data type, across 10 training epochs.

### B.3 DETAILS OF ENTITY EXTRACTION EVALUATION

To assess the reliability of entity extraction process, we conducted evaluation on outputs from the LLaMA3-Instruct-8B model using the CNN/DailyMail and SQuAD v2 datasets. For each dataset, we randomly sampled 50 cases and compared the extracted entities with two sets of references:

---

[2]https://github.com/hiyouga/LLaMA-Factory

DeepSeek R1 results and human annotations. Entity extraction performance was measured using F1 score, where an entity was considered correct if successfully identified, regardless of its type. This lenient criterion led to high F1 scores—CNN/DailyMail: 90.49% (DeepSeek R1) vs. 87.62% (human); SQuAD v2: 89.54% (DeepSeek R1) vs. 87.11% (human). This evaluation demonstrates that relying solely on the LLM's capabilities enables effective entity extraction, providing a solid foundation for subsequent steps.

# C   INTRODUCTION TO THE PROMPT USED

## C.1   INSTRUCTION DESCRIPTION FOR LLMINERTIA

### C.1.1   ENTITY EXTRACTION

Given a text input, we prompt the LLM to extract all entities. The instruction is as follows:

> **Instruction:** Carefully read the following input and list all important entities, in the format below.
> - Output entities as a comma-separated list. No extra words.
>
> **Example:**
> Input: The patient reports experiencing a persistent cough and fever for three days. Chest X-ray was performed with no abnormalities detected.
> Output: Cough, Fever, three days, Chest X-ray
>
> **Input:** [Your Input]
> **Output:**

### C.1.2   CO-OCCURRENCE PROBE

For each extracted entity, we prompt the LLM to output the three most closely related entities, in the format of co-occurrence pairs. The instruction is as follows:

> **Instruction:** For each given entity, output three entities that are most closely related to it. List all results in the form of tuple. See the format and example below.
>
> **Format:**
> Entities: Entity1, Entity2
> Output:
> (Entity1, Co1), (Entity1, Co2), (Entity1, Co3)
> (Entity2, Co1), (Entity2, Co2), (Entity2, Co3)
>
> **Example:**
> Entities: Headache, Cough
> Output:
> (Headache, Dizziness), (Headache, Nausea), (Headache, Chest pain)
> (Cough, Shortness of breath), (Cough, Sore throat), (Cough, Fatigue)
>
> **Entities**:[Your Entities]
> **Output:**

### C.1.3   COUNTER-INERTIAL REASONING

Finally, we prompt the LLM to answer the given question, explicitly encouraging evidence-grounded reasoning with the adaptive reminder inserted. The response format is strictly enforced. The instruction is as follows:

> **Instruction:** Please answer the following question and strictly respond in the format: [Conclusion]: Yes/No.
>
> **Here are two examples:**
>
> *Example 1:*
> Question: The patient's main complaint during this visit is Headache, which can sometimes indicate Hypertension, although no clinical or laboratory abnormalities related to Hypertension have been found. Does Hypertension exist or hold?

Reminder: Headache and Hypertension may not necessarily be related. Please carefully examine the original text and make a thorough judgment.
Answer: [Conclusion] No.

*Example 2:*
Question: The patient has been experiencing persistent cough for several days, which can sometimes suggest Pneumonia; imaging results showed clear lungs with signs of infection Pneumonia. Does Pneumonia exist or hold?
Reminder: Cough and Pneumonia may not necessarily be related. Please carefully examine the original text and make a thorough judgment.
Answer: [Conclusion] Yes.

**Question:** [Your question]
**Reminder:** [Adaptive reminder]
**Answer:**

## C.2 THE PROMPT FOR THE BASE AND PROMPT VERSIONS USED FOR MODEL TESTING

Table 6: For the summary and QA tasks, show the instructions for the base and prompt versions of the three model bases.

| Tasks | Base | Prompt |
|---|---|---|
| Summary | Write a summary of the following news. | Write a summary of the following news. Attention should be paid to the faithfulness of the abstract with the original text to avoid generating content with hallucinations. |
| QA | You are a question answerer. You should answer the questions directly based on the given reference without adding any prefixes or suffixes, and without analyzing the answers. After answering the question, do not say anything else. Reference document: ...... Please answer the question based on the above reference: | You are a question answerer. You should answer the questions directly based on the given reference without adding any prefixes or suffixes, and without analyzing the answers. After answering the question, do not say anything else. Please do not output content that is inconsistent with the context, and avoid giving irrelevant or contradictory answers. Reference document: ...... Please answer the question based on the above reference: |

## D   THE USE OF LARGE LANGUAGE MODELS

Large language models (LLMs) were utilized in the preparation of this paper solely for grammar checking and minor language polishing.

