# OpenReview forum: "LLMInertia: Investigating and Mitigating Large Language Models' Unfaithfulness to Input Evidence from a Cognitive Inertia Perspective"
_ICLR.cc/2026/Conference — Submitted to ICLR 2026_

### Official Review · Reviewer_MXRp · 2025-10-29

**Soundness:** 2
**Presentation:** 3
**Contribution:** 2
**Rating:** 4
**Confidence:** 3

**Summary:**

This paper investigates a fundamental source of hallucination in large language models (LLMs), termed cognitive inertia—the tendency of models to rely excessively on learned co-occurrence patterns even when explicit evidence contradicts them. The authors systematically demonstrate that as co-occurrence strength in pretraining data increases, model faithfulness to input evidence decreases. To address this issue, they propose an adaptive counter-inertial reasoning framework (LLMInertia), which detects potential cognitive inertia in a given input and injects the dynamically generated counter-inertial reminders into the prompt to encourage evidence-based reasoning. Experimental results on controlled co-occurrence induction datasets show notable improvements. Further evaluations across multiple summarization and question-answering benchmarks, using three LLM backbones, confirm the method’s robustness and general effectiveness.

**Strengths:**

1. The paper studies LLM hallucination, a highly valuable research problem that is crucial for ensuring application safety and reliability.

2. The preliminary experiments are interesting and informative, effectively illustrating the issue of Cognitive Inertia in LLMs.

3. The proposed hallucination mitigation approach is simple yet effective, and demonstrates consistent accuracy improvements across different model families such as Qwen and Llama.

**Weaknesses:**

1. The causes of context hallucination (i.e., LLM unfaithfulness to explicit input evidence) are multifaceted and cannot be fully attributed to cognitive inertia. For example, context length is also a key factor. When the input becomes longer, models may fail due to attention dispersion. I therefore recommend that the authors include additional statistics to isolate the specific impact of cognitive inertia on hallucination. Moreover, when the input evidence conflicts with the knowledge learned during pretraining, LLMs sometimes generate responses opposite to the input.

2. The proposed hallucination mitigation method may not be fully reliable. The entity extraction and probing processes depend heavily on both the LLM’s own capabilities and the quality of the external knowledge base (Co-oc set), which could accumulate noise. In addition, injecting counter-inertial reminders into the prompt does not guarantee that the model will correctly detect and resist cognitive inertia—this may require solutions beyond prompt engineering.

3. The preliminary experiments might not be strictly necessary. As noted by the authors themselves (Lines 318–319), the influence of high-frequency co-occurrence entities on LLM reasoning is already a commonly recognized phenomenon.

**Questions:**

1. In Table 1, the performance of the three models under Halu Eval appears comparable to that of SFT—some better, some worse. Could the authors provide further explanation for this inconsistency?

---

> ### Author Response · Authors · 2025-11-22
> **Rebuttal by Authors (1/2)**
>
> We are very grateful for your kind and constructive feedback. Below are our responses to each of the concerns you raised.
>
> **1. Discussing the Specific Impact of Cognitive Inertia on Hallucination.** Thank you very much for your insightful feedback. We fully agree that context hallucinations can arise from multiple factors, including context length and attention dispersion. In this work, we specifically focus on cognitive inertia—a key contributor characterized by the model’s tendency to rely on learned co-occurrence associations instead of input evidence. Following the reviewer’s suggestion, we isolate and quantify its prevalence and impact through targeted analysis.
>
> Specifically, we sampled 100 low-AlignScore outputs from the Base model (LLaMA-3-Instruct-8B) for both the CNN/DailyMail and SAMSum datasets, and manually annotated the error types. Among all hallucination errors, those driven by cognitive inertia—i.e., over-reliance on co-occurrence associations—account for **31.08% in CNN/DailyMail** and **33.79% in SAMSum**, indicating that hallucinations from cognitive inertia in LLMs are frequent and a major contributor to overall hallucinations.
>
> We further examined how well all baselines and LLMInertia mitigate inertia-driven errors. The table below shows the mitigation ratios—the percentage of such hallucinations in the Base model corrected by each method.
>
> **Table. Mitigation Ratios of Inertia-Driven Hallucinations**
> | Method| CNN/DailyMail | SAMSum |
> |----|------|---------|
> | Prompt | 22.86% | 22.58% |
> | SFT | 27.14%| 25.81% |
> | SymbCoT| 22.86% | 20.97%  |
> | Lookback| 20.00% |19.35% |
> | **LLMInertia** | **34.29%**| **32.26%** |
>
> As shown, all methods mitigate cognitive inertia-induced hallucinations to varying degrees, with lower mitigation ratios on the more challenging SAMSum dialogue summarization dataset than on the standard CNN/DailyMail summarization dataset. Importantly, LLMInertia consistently achieves the highest mitigation ratios on both datasets, outperforming all baselines. These results highlight the effectiveness of LLMInertia’s adaptive reminder mechanism in combating over-co-occurrence bias and encouraging the model to focus on input evidence. We will incorporate this important experimental analysis into the revised manuscript.
>
> **2. Discussion of LLMInertia Details.** Thank you for your thoughtful comments. To clarify, entity extraction and co-occurrence probing in our method do not rely on any external knowledge bases; both processes are implemented through direct instruction calls to the target LLM. To assess the reliability of entity extraction, we evaluated outputs from LLaMA3-Instruct-8B on the CNN/DailyMail and SQuAD v2 datasets, comparing DeepSeek R1 and human annotations for 50 randomly sampled cases per dataset. The strong F1 scores (**CNN/DailyMail: 90.49% vs. 87.62%; SQuAD v2: 89.54% vs. 87.11%**) are primarily attributed to our requirement for accurate entity identification rather than fine-grained classification, which provides a solid foundation for subsequent steps in the pipeline.
>
> The co-occurrence probing step leverages the LLM’s capability to recall common entity associations from training, as supported by previous work[1,2]. To mitigate noise, we retain only those entity pairs actually present in the input context, minimizing irrelevant reminders.
>
> Distinct from standard prompt engineering, our method adaptively generates context-specific counter-inertial reminders to guide the LLM’s attention toward error-prone associations. As shown in the table below, LLMInertia delivers consistent improvements in both consistency and faithfulness compared to standard prompt-based methods, averaged over three backbone models.
>
> | Methods | CNN/Daily Mail (Consistency) | CNN/Daily Mail (AlignScore) | SAMSum (Consistency) | SAMSum (AlignScore) | SQuAD V2 (Anah-v2) | SQuAD V2 (AlignScore) | HaluEval (Anah-v2) | HaluEval (AlignScore) |
> |---|---|--|--|---|--|--|---|--|
> | LLMInertia vs Prompt (avg. gain) | +3.15% | +1.14%  | +0.97% | +1.91% | +2.76%  | +1.09% | +1.81% | +0.45%   |
>
>
> In summary, we recognize that mitigating hallucinations and cognitive inertia in LLMs is a fundamentally challenging task. Our approach addresses this challenge by decomposing the problem into tractable steps that are well aligned with the LLM’s capabilities. As demonstrated by the experimental results above, each step maintains high quality and introduces minimal noise, ensuring that overall performance remains robust and reliable. Nevertheless, we acknowledge that further improvements may require more sophisticated intervention mechanisms or learning-based strategies, and we will explore these directions in future work.
>
> Thank you again for your constructive feedback
>
> > [1] Zechen Sun et al. Exploring and mitigating shortcut learning for generative LLMs. LREC-COLING 2024.
> >
> > [2] Xiaoyu Liu et al. Large language models and causal inference: A comprehensive survey. NAACL 2025.

---

> ### Author Response · Authors · 2025-11-22
> **Rebuttal by Authors (2/2)**
>
> **3. Discussion of Section 3 Experiments.** Thank you for your thoughtful comments. While prior studies have recognized the influence of high-frequency co-occurring entities on LLM reasoning, our experiments in Section 3 are designed to move beyond preliminary observations and to form an important part of our contribution. Specifically, we empirically reveal the cognitive inertia in LLMs—the tendency to overly rely on co-occurrence associations in pretraining even in the face of new or contradictory input evidence—as a key contributing factor to hallucinations, which has not previously been identified.
>
> The statement in Lines 318–319, regarding prior work showing that LLMs are likely to generate high-frequency co-occurring entities, serves as background to motivate the design of our Cognitive Inertia Probe within the LLMInertia framework (Section 4.1), and is not the core novelty of our work.
>
> Our study contributes to the field by:
>
> >**Establishing Causality:** We designed interventional experiments by systematically manipulating co-occurrence strength (via data proportion and training epochs), allowing us to isolate the specific impact of co-occurrence associations on hallucinations unfaithful to input evidence.
> >
>
> >**Quantifying the Effect:** We explicitly quantified the phenomenon using the hallucination induction rate, demonstrating that increased co-occurrence strength mechanistically amplifies the model's unfaithfulness to explicit input evidence.
>
>
> >**Revealing the Mechanism:** Our findings uncover cognitive inertia in LLMs as a key mechanism underlying unfaithful outputs—an effect not previously established in the literature.
>
> We hope these clarifications highlight the value and originality of our experiments, and we appreciate the opportunity to elaborate on their significance.
>
> **4.Discussion on Performance on the HaluEval Dataset.** Thank you for your thoughtful question. We believe the comparable—and sometimes even slightly superior—performance of SFT on HaluEval can be attributed to both the characteristics of the dataset and the properties of the methods being compared.
>
> First, HaluEval is constructed using a two-stage sampling-then-filtering process, resulting in hallucination samples that are highly similar to real data, with differences concentrated in key factual spans. These samples are generated based on a limited set of specific patterns, making the overall structure of the dataset more predictable. This leads to a relatively low difficulty level for the evaluated models, and therefore narrows the performance gap between different approaches. Moreover, as a supervised training method, SFT tends to perform particularly well when evaluation datasets—such as HaluEval—exhibit structured and recurrent patterns that are similar to its training data. Due to these two reasons, we observe that SFT can sometimes even slightly outperform our method under certain backbone models or metrics, making the improvement brought by our approach less consistent on this dataset.
>
> In contrast, datasets like SAMSum (which is conversational, multi-speaker, and less structured) and SQuAD v2.0 (which features a larger variety of question types, including unanswerable questions) are more challenging and diverse. In these cases, the generalization and robustness advantages of our training-free, adaptive method become more pronounced, as reflected in the consistently better performance across these more challenging scenarios. We will incorporate this important analysis into the revised manuscript. Thank you again for the insightful comments.
>
> We are especially grateful for your recognition of our work and for the insightful suggestions you have shared. Should you have any additional questions or further recommendations, please feel free to let us know.

---

> ### Author Response · Authors · 2025-11-26
> **Follow-Up on Review and Feedback**
>
> Dear Reviewer MXRp,
>
> We hope this message finds you well.
>
> We have thoroughly addressed all of your questions and concerns, including conducting the additional experiments you requested, and have provided detailed responses in the rebuttal.
>
> As the rebuttal deadline approaches, we would greatly appreciate it if you could share your updated thoughts based on the rebuttal, or do not hesitate to let us know if you have additional questions, and we will respond promptly.
>
> Thank you again for your thoughtful review and for your valuable contributions to improving the quality of our paper.
>
> Kind regards,
>
> Paper 8867 Authors

---

### Official Review · Reviewer_aoiE · 2025-10-30

**Soundness:** 3
**Presentation:** 2
**Contribution:** 2
**Rating:** 2
**Confidence:** 4

**Summary:**

The paper investigates cognitive inertia in LLMs. The authors show that increasing co-occurrence frequency during training amplifies unfaithful, hallucinated outputs. Building on this insight, they propose LLMInertia, an adaptive counter-inertial reasoning framework that probes the model to identify potential co-occurrence biases and injects counter-inertial reminders into the prompt to encourage evidence-based reasoning.

**Strengths:**

1.	The paper introduces cognitive inertia as a unifying explanation for LLMs’ unfaithfulness to input evidence, framing it analogously to human inertia in reasoning.
2.	The proposed LLMInertia framework is computationally efficient. It requires no retraining and shows measurable improvements (≈ 14% hallucination reduction, ≈ 12% accuracy gain).
3.	Results on multiple datasets and against strong baselines (Prompting, SymbCoT, Lookback, SFT) indicate robustness and generalizability.

**Weaknesses:**

1. Prior works such as [1] and [2] already discuss the influence of co-occurrence and the interplay between parametric and contextual knowledge. The present paper could better position itself by clarifying whether cognitive inertia is fundamentally distinct or simply a reframing of these effects.
[1] Kang, Cheongwoong, and Jaesik Choi. "Impact of co-occurrence on factual knowledge of large language models." arXiv preprint arXiv:2310.08256 (2023).
[2] Cheng, Sitao, et al. "Understanding the interplay between parametric and contextual knowledge for large language models." arXiv preprint arXiv:2410.08414 (2024).

2. While demonstrated to be effective, the proposed method works on entity-level questions. The proposed method cannot generalize to open-ended reasoning or multi-hop inference tasks where evidence is implicit, which is a big limitation.


3. The paper claims the medical pairs to build the co-occurrence induction training corpus were selected by statistically identifying high-frequency co-occurrences in the MIMIC dataset. What’s the exact process to choose these pairs? How does the author vary co-occurrence ratios to build the dataset? Those details are important for reproducibility and evaluation.

**Questions:**

See weakness.

---

> ### Author Response · Authors · 2025-11-22
> **Rebuttal by Authors (1/2)**
>
> We sincerely appreciate your constructive feedback. Below, we address each of the concerns you raised.
>
> **1. Difference from Co-occurrence-based Methods.** Thank you for raising this important question. While previous studies discuss the influence of co-occurrence bias and the interplay between parametric and contextual knowledge, they do not establish a causal connection between excessive co-occurrence in pretraining and LLMs’ unfaithful or hallucinated responses to input evidence. This important connection cannot be directly inferred from prior work, but is instead demonstrated through our targeted experimental interventions and quantitative analysis. Our work reveals the phenomenon of cognitive inertia in LLMs, identifies it as a key cause of unfaithful hallucinations, and further develops an effective mitigation mechanism—rather than offering a simple reframing of existing work. To clarify these distinctions, we first briefly outline the focus of each work, and then provide a summary table highlighting the core differences.
>
> **LLMInertia:**  We identify cognitive inertia in LLMs——the tendency to overly rely on co-occurrence associations even when confronted with new or contradictory input evidence—as an important contributing factor to unfaithfulness hallucinations. Through targeted experiments, we show that LLM adherence to explicit input evidence decreases as the strength of co-occurrence associations in pretraining data increases. To address this, we propose a training-free, adaptive framework that probes the model to identify high-frequency co-occurrence associations and injects counter-inertial reminders into the prompt to encourage evidence-based reasoning.
>
> **[1] Kang & Choi (2023):**  Kang & Choi et al. study the impact of co-occurrence on factual knowledge recall in LLMs. Their key finding is the co-occurrence bias, where LLMs tend to prefer frequently co-occurred subject-object pairs over the correct answer, which leads to particular difficulty in recalling rare facts. Their mitigation strategy relies on data-level debiasing via undersampling and finetuning.
>
> **[2] Cheng et al. (2024):**  Cheng et al. examine how LLMs integrate internal parametric knowledge with external contextual knowledge. Through the ECHOQA benchmark, they find that LLMs frequently suppress their internal knowledge when contextual cues are present, even when both sources are complementary. While targeted prompting (such as step-by-step reasoning) offers partial mitigation, it does not fully address the underlying limitation of reliably combining these knowledge sources.
>
> The following table summarizes the fundamental differences across these works:
>
> | Dimension                         | [1] Kang & Choi (2023)                                                                                                                                 | [2] Cheng et al. (2024)                                                                             | **LLMInertia**                                                                                                   |
> |-----------------------------------|-------------------|---------------------------------|--------------------------------------------------|
> | **Main Problem Addressed**        | LLM exhibits failures in recalling factual knowledge                                                             | LLM exhibits suppression of internal parametric knowledge when external contextual knowledge is present          | LLM exhibits unfaithfulness hallucination to explicit input evidence                          |
> | **Mechanistic Cause/Discovery**   | Empirically analyzes LLM failures in recalling facts whose subject and object rarely co-occur in pretraining data.  | Observes that LLMs tend to use external contextual knowledge over internal parametric knowledge when both are present. | Designs controlled interventions by varying the strength of co-occurrence associations in pretraining. Quantitatively measures hallucination induction rate, and causally demonstrates that increasing co-occurrence strength leads to diminished adherence to explicit input evidence. Reveals cognitive inertia in LLMs as a key reason underlying unfaithful outputs. |
> | **Solution Method**               | Data-level debiasing via undersampling and finetuning                                                                                       | Prompt- and instruction-based interventions                                                          | A training-free framework that probes cognitive inertia in LLMs for each input and injects adaptive counter-inertial reminders into the prompt to encourage evidence-based reasoning.
>
> We sincerely hope that the above comparative discussion will assist you in recognizing the originality and value of our work. Furthermore, we have respectfully included these two papers as important references in the revised manuscript in response to your helpful suggestions.

---

> ### Author Response · Authors · 2025-11-22
> **Rebuttal by Authors (2/2)**
>
> **2. Discussion on More Complex Inference Tasks.** Thank you for your valuable comments regarding the applicability of our method to more complex settings.
>
> To better understand the practical impact of cognitive inertia-driven hallucinations, we sampled 100 low AlignScore outputs from the base model (LLaMA-3-Instruct-8B) on the CNN/DailyMail and SAMSum datasets, and manually annotated error types. Our analysis shows that, among all hallucination errors, those driven by cognitive inertia—i.e., over-reliance on co-occurrence associations—account for 31.08% in CNN/DailyMail and 33.79% in SAMSum. This indicates that such errors are frequent and a major contributor to overall hallucinations. Furthermore, the consequences of the cognitive inertia-driven hallucinations can be severe in high-stakes domains, potentially resulting in patient harm in medicine or misjudgments in the legal field—where grounding in evidence is crucial.
>
> We also compared several mainstream mitigation methods on these subsets. As shown in the table below, LLMInertia consistently achieved the highest mitigation ratios and outperformed all baseline approaches. These results underscore not only the prevalence and practical significance of inertia-driven hallucinations, but also the superior effectiveness of our proposed approach.
>
> Table. Mitigation Ratios of Inertia-Driven Hallucinations
> | Method      | CNN/DailyMail | SAMSum   |
> |-------------|---------------|----------|
> | Prompt      | 22.86%        | 22.58%   |
> | SFT         | 27.14%        | 25.81%   |
> | SymbCoT     | 22.86%        | 20.97%   |
> | Lookback    | 20.00%        | 19.35%   |
> | **LLMInertia** | **34.29%**   | **32.26%** |
>
> We fully acknowledge that generalizing to multi-hop reasoning and implicit evidence scenarios remains a significant and open challenge beyond our current scope. In this work, we focus on frequent and consequential hallucinations—specifically, those arising from explicit co-occurrence associations. Nonetheless, the LLMInertia framework is designed to be extensible. For instance, the probing component can be enhanced with graph-based methods or more sophisticated bias detection techniques, enabling our approach to address more complex cases such as multi-hop reasoning in the future. We will clarify this potential and highlight it as an important direction in our revised manuscript.
>
> **3. Experimental Setup Details.** Thank you for your question regarding the selection of medical entity pairs and control of co-occurrence ratios.
>
> **1) Selection of Medical Entity Pairs:** We used the Bio_ClinicalBERT model [1] to extract clinical entities from the MIMIC dataset and computed sentence-level co-occurrence frequencies for all entity pairs. The top 80 most frequent pairs were initially selected. Medical professionals then reviewed these pairs to ensure each represented a genuine clinical association rather than a direct causal relation. After expert validation, 40 high-frequency pairs were retained for the final corpus. The full list of validated pairs is included in Appendix A.1 to ensure transparency and reproducibility.
>
> **2) Design and Variation of Co-occurrence Ratios:** To systematically examine two key effects—(1) how excessive co-occurrence data and (2) how extensive pretraining under a fixed co-occurrence ratio influence faithfulness hallucinations—we manipulated co-occurrence frequency in two ways:
> > a. We created several training corpora with varying proportions of co-occurrence data (25%, 50%, 75%, and 100%). To keep the dataset size consistent, the remainder was filled with Wiki-en text [2] as needed.
> >
>
> >b. We fixed the co-occurrence ratio at 25% and varied the number of training epochs, thus increasing exposure to the co-occurrence pairs and allowing us to measure cumulative effects.
>
> All processes, training details, and implementation parameters are documented in Appendix B to facilitate reproducibility. We hope this provides clarity on our methodology and demonstrates its transparency.
>
> [1] Alsentzer E, Murphy J, Boag W, et al. Publicly available clinical BERT embeddings. Proceedings of the 2nd Clinical NLP Workshop, 2019: 72-78.
> [2] Mandy Guo, Zihang Dai, Denny Vrandečić, and Rami Al-Rfou. Wiki-40B: Multilingual language model dataset. Proceedings of LREC, 2020: 2440–2452.
>
>
> We are deeply grateful for your thorough and meticulous evaluation, and we truly appreciate the time and effort you have devoted to reviewing our manuscript. Your valuable feedback has significantly contributed to improving the completeness of our work. If you have any further questions or would like additional clarification, please feel free to let us know.

---

> ### Author Response · Authors · 2025-11-26
> **Follow-Up on Review and Feedback**
>
> Dear Reviewer aoiE,
>
> We hope this message finds you well.
>
> We have carefully addressed all your questions and concerns, including conducting additional experiments as requested, and have provided detailed responses in the rebuttal.
>
> As the rebuttal deadline is approaching, we would deeply appreciate it if you could share your updated thoughts based on the rebuttal, or do not hesitate to let us know if you have additional questions, and we will respond promptly.
>
> Thank you again for your thoughtful review and your invaluable contributions to the quality of this paper.
>
> Kind regards,
>
> Paper 8867 Authors

---

### Official Review · Reviewer_R31P · 2025-11-01

**Soundness:** 2
**Presentation:** 2
**Contribution:** 2
**Rating:** 6
**Confidence:** 3

**Summary:**

This paper focuses on understanding the recent observations on the cognitive inertia in LLMs -- the tendancy to overly rely on the pattern in the training data, instead of focusing on new or contradictory evidence provided in the input context.

The authors first provide empirical results showing correlation between the strength of co-occurence association in the pre-training data and the adherence of LLMs to explicit input evidence (which might contradict with the pre-training data).

This paper proposes an adaptive counter-inertial reasoning framework, which let an LLM generates counter-inertial reminder and another LLM generates answers based on the first LLMs.

**Strengths:**

This paper focuses on an interesting research question.
The empirical observation on the "cognitive inertia" makes sense.

**Weaknesses:**

The idea for the hallucination mitigation problem is not that clear.

**Questions:**

None

---

> ### Author Response · Authors · 2025-11-22
>
> Thank you very much for your positive feedback, especially your recognition of our interesting research question and the validity of our empirical observations on "cognitive inertia." We appreciate your concern regarding the clarity of our hallucination mitigation approach and provide clarification below.
>
> Our approach is based on the phenomenon of cognitive inertia in LLMs, where models tend to rely on strong pre-trained co-occurrence associations even when these conflict with explicit input evidence, which can result in unfaithful outputs. Inspired by human counter-inertial reasoning, we propose an adaptive counter-inertial reasoning framework that guides the model to overcome these biases and focus on the actual evidence provided in the input.
>
> Specifically, the framework works as follows:
>
> > **Self-probing for cognitive inertia:** We first extract all entities from the input using an extraction function. For each extracted entity, we apply a probing function to uncover its most frequent co-occurring entities in the LLM’s internal memory, forming adaptive co-occurrence association pairs relevant to the given input.
> >
>
> >**Generation of adaptive reminders:** Based on these identified associations, the model automatically generates a tailored "counter-inertial reminder" (for example, "entity a and entity b may not necessarily be related. Please examine the original
> text carefully and make a thorough judgment).
> >
>
> >**Prompt injection for guided reasoning:** The reminder is injected into the model’s prompt along with the context and question, guiding the model to focus more on input evidence during reasoning and thereby reducing faithfulness hallucination.
>
> The functions of entity extraction, co-occurrence probing, and evidence injection described above are all implemented by prompting the target LLM (with full prompt details provided in Appendix C). We hope that this detailed explanation addresses any uncertainties regarding our hallucination mitigation approach.
>
> Thank you again for your feedback. We welcome any additional comments or suggestions you may have.

---

### Official Review · Reviewer_BRzS · 2025-11-02

**Soundness:** 3
**Presentation:** 3
**Contribution:** 3
**Rating:** 6
**Confidence:** 3

**Summary:**

This paper investigates how co-occurrence frequency during pretraining contributes to faithful hallucination. They create synthetic training and test data with adjustable co-occurrence frequencies, and their co-occurrence induction experiments show that hallucination induction rate increases as LLMs are exposed to more co-occurrence associations during pretraining. They define LLM’s tendency to overly rely on co-occurrence associations as cognitive inertia. Based on this observation, they propose a prompting framework, named LLMInertia, to promote faithful output by identifying high-frequency co-occurring associations in the input context and including instructions to mitigate cognitive inertia. They evaluate their prompting framework on co-occurrence induction data, summarisation and question answering datasets to show that LLMInertia significantly reduces hallucination induction rate and improves accuracy.

**Strengths:**

- They conduct a detailed analysis of the impact of co-occurrence frequency on LLM generation, providing new insights into the cause of faithful hallucination
- They propose a simple and effective prompting method to improve the faithfulness of the generated outputs
- The paper is well structured and easy to follow

**Weaknesses:**

The experiment setup and analysis for the evaluation on question answering and summarisation benchmarks requires more clarification.

**Questions:**

I have some questions about the experiments on question answering and summarisation datasets.

- What is the difference between base model and naive prompting baselines? Also, it would be helpful to include a CoT baseline to better contextualise your results
- Factual consistency metrics such as AlignScore can be affected by the output length. Does the LLM tend to generate longer output with LLMInertia method?
- It is not clear how LLM reasoning is affected by the counter-inertial reminder from the case study. It would be better to include some reasoning steps or full generation outputs for clarity.

---

> ### Author Response · Authors · 2025-11-22
> **Rebuttal by Authors (1/3)**
>
> We sincerely appreciate your constructive feedback and recognition of our work. Regarding your overall comment about the need for more clarification in our experimental setup and analysis, we have addressed each of your specific questions below and will incorporate clarifications into **Section 4.3 and Appendix C.2** of the revised manuscript to improve clarity.
>
> **1. Clarification of Base Models and Prompts.** Thank you for your valuable questions. We apologize for any confusion caused by our experimental setup and appreciate the opportunity to clarify it here.
>
> We evaluated three competitive large language models—LLaMA-3-8B [1], Qwen2.5-7B [2], and LLaMA-3-70B [1]—all of which are publicly released instruct-tuned versions. To clarify the difference between the Base and Naive Prompting settings, we now include the following table and will add it to the appendix in the final version. As shown in Table 1 below, the base setting uses only the minimal task instruction, while naive prompting augments this with an explicit request for faithfulness.
>
> **Table 1: For the summary and QA tasks, show the instructions for the base and prompt versions of the three model bases.**
> | Tasks   | Base | Prompt |
> |---------|------|--------|
> | Summary | Write a summary of the following news. | Write a summary of the following news. **Attention should be paid to the faithfulness of the abstract with the original text to avoid generating content with hallucinations.** |
> | QA      | You are a question answerer. You should answer the questions directly based on the given reference without adding any prefixes or suffixes, and without analyzing the answers. After answering the question, do not say anything else. Reference document: ...... Please answer the question based on the above reference. | You are a question answerer. You should answer the questions directly based on the given reference without adding any prefixes or suffixes, and without analyzing the answers. After answering the question, do not say anything else. **Please do not output content that is inconsistent with the context, and avoid giving irrelevant or contradictory answers.** Reference document: ...... Please answer the question based on the above reference. |
>
> > [1] Abhimanyu Dubey, Abhinav Jauhri, Abhinav Pandey, Abhishek Kadian, Ahmad Al-Dahle, Aiesha Letman, Akhil Mathur, Alan Schelten, Amy Yang, Angela Fan, et al. The llama 3 herd of models. arXiv preprint arXiv:2407.21783, 2024.
> >
> > [2] An Yang, Baosong Yang, Beichen Zhang, Binyuan Hui, Bo Zheng, Bowen Yu, Chengyuan Li, Dayiheng Liu, Fei Huang, Haoran Wei, et al. Qwen2. 5 technical report. arXiv preprint arXiv:2412.15115, 2024a.

---

> ### Author Response · Authors · 2025-11-22
> **Rebuttal by Authors (2/3)**
>
> **2.Inclusion of CoT Baseline Results.** Thank you for your suggestion. Our original submission reported results for SymbCoT [1], a recent faithfulness-oriented CoT method. In response, we have conducted additional experiments with standard CoT prompting [2], with results shown below (to be included in the revised manuscript).
>
> Briefly, CoT and SymbCoT baselines perform similarly, with SymbCoT slightly ahead of CoT. LLMInertia achieves a clear boost, surpassing CoT by 2.52% and SymbCoT by 2.49% on Qwen-2.5-7B across all tasks and metrics, with similar improvements on other model bases. These gains stem from LLMInertia’s adaptive mitigation of cognitive inertia during inference, resulting in improved results.
>
> | Methods          | CNN/Daily Mail (Consistency(↑)) | CNN/Daily Mail (AlignScore(↑)) | SAMSum (Consistency(↑)) | SAMSum (AlignScore(↑)) | SQuAD V2 (Anah-v2(↓)) | SQuAD V2 (AlignScore(↑)) | HaluEval (Anah-v2(↓)) | HaluEval (AlignScore(↑)) |
> |--------------------------|-------------------------------|-------------------------------|------------------------|------------------------|------------------------|--------------------------|-----------------------|-------------------------|
> |      |                               |                               |                        |          **LLaMA-3-8B**               |                        |                          |                       |                         |
> | Base                     | 86.4                          | 83.28                         | 90.79                  | 90.67                  | 14.32                  | 95.94                    | 12.48                 | 96.35                   |
> | CoT                      | 86.47                         | 84.26                         | 90.14                  | 88.41                  | 13.54                  | 95.46                    | 11.30                 | 96.47                   |
> | SymbCoT                  | 86.52                         | 84.33                         | 90.23                  | 88.37                  | 13.7                   | 95.57                    | 11.66                 | **96.58**                   |
> | **LLMInertia**               | **90.33**                     | **88.24**                     | **91.54**              | **92.33**              | **12.53**              | **96.34**                 | **8.21**              | 96.35               |
> |      |                               |                               |                        |            **LLaMA-3-70B**             |                        |                          |                       |                         |
> | Base                     | 89.05                         | 85.41                         | 92.37                  | 91.43                  | 11.52                  | 96.83                    | 10.32                 | 96.77                   |
> | CoT                      | 89.02                         | 85.49                         | 90.97                  | 91.43                  | 11.15                  | 96.13                    | 10.06                 | 96.65                   |
> | SymbCoT                  | 89.06                         | 85.57                         | 91.04                  | 91.52                  | 11.38                  | 96.47                    | 10.23                 | 96.76                   |
> | **LLMInertia**               | **91.4**                      | **86.75**                     | **93.02**              | **91.97**              | **9.56**               | **97.04**                 | **8.33**              | **97.22**               |
> |      |                               |                               |                     |              **Qwen-2.5-7B**              |                        |                          |                       |                         |
> | Base                     | 83.76          | 82.56           | 85.23             | 86.42                  | 17.22                  | 93.65                    | 14.75                 | 95.05                   |
> | CoT                      | 84.34             | 81.97         | 84.22          | 86.71       | 13.06                  | 93.58                    | 11.23                 | 95.33                   |
> | SymbCoT                  | 84.45                         | 81.84       | 84.31                  | 86.72                  | 13.25                  | 93.67                    | 11.38                 | 95.35                   |
> | **LLMInertia**               | **89.31**                     | **85.4**           | **88.03**       | **89.14**              | **11.22**              | **95.56**                 | **10.47**             | **96.1**                |
>
> > [1] Xu J, Fei H, Pan L, et al. Faithful Logical Reasoning via Symbolic Chain-of-Thought. ACL 2024.
> >
> > [2] Wei J, Wang X, et al. Chain-of-thought prompting elicits reasoning in large language models. NeurIPS 2022, pp. 24824–24837.

---

> ### Author Response · Authors · 2025-11-22
> **Rebuttal by Authors (3/3)**
>
> **3. Analysis of Output Length.** Thank you for your insightful comment. In response, we conducted a statistical analysis of output lengths across all methods on the CNN/Daily Mail dataset using the LLaMA-3-Instruct-8B backbone.
>
> The results (see table below) indicate that LLMInertia’s output length is closely aligned with other baselines in terms of mean, median, and standard deviation. Therefore, its performance gains are not attributable to generating longer outputs. Instead, these improvements arise from counter-inertial reminders that shift the model’s attention from spurious co-occurrence-based associations to critical evidence present in the input, thereby enhancing factual consistency.
>
> **Table: Statistical analysis of output lengths for all methods on CNN/Daily Mail dataset**
>
> | Method     | Mean | Median | Std Dev |
> |------------|------|--------|---------|
> | Base       | 117  | 109    | 38      |
> | Prompt     | 119  | 110    | 43      |
> | SFT        | 118  | 108    | 39      |
> | CoT    | 121  | 111    | 39      |
> | SymbCoT    | 119  | 109    | 43      |
> | Lookback   | 122  | 111    | 41      |
> | LLMInertia | 120  | 110    | 42      |
>
> **4.Case Study Completion.**
> Thank you very much for your constructive suggestion. In the revised manuscript, we will supplement both case study examples with key reasoning processes. As an illustration, below we present the completed reasoning steps for the first example from the Co-occurrence induction dataset:
>
> >**Context:** The current chief complaint is stomach pain, and no clinical features related to gastric hemorrhage being present...
> >
> >**Question:** Does Gastric Hemorrhage exist or hold?
> >
> >**Adaptive Reminder by LLMIntertia:** Stomach pain and gastric hemorrhage may not necessarily be related. Please carefully examine the original text and make a thorough judgment.
> >
> >**Without LLMIntertia:** ...Based on typical clinical cases, stomach pain often leads to concerns about more severe issues such as bleeding in the stomach....[Conclusion] Yes (incorrect answer).
> >
> >**With LLMIntertia:** ...The reminder highlights that not all cases of stomach pain are caused by bleeding in the stomach. Since there is no direct evidence of gastric hemorrhage, I conclude that gastric hemorrhage does not exist for this patient...[Conclusion] No (correct answer).
>
>
> As suggested by the reviewer, showing the key reasoning steps before and after applying LLMInertia indeed more clearly highlights the positive impact of LLMInertia on the model’s faithful reasoning. This comparison of the reasoning process demonstrates how the counter-inertial reminder shifts the model’s reasoning from superficial associations to evidence-based judgment.
>
> Thank you again for your helpful feedback, which has significantly improved the clarity and transparency of our work.

---

### Author Response · Authors · 2025-11-28
**General Responses**

We thank all reviewers for their detailed feedback and constructive suggestions on multiple aspects of our work. We appreciate the recognition of the originality and design of LLMInertia—a training-free, adaptive framework that investigates and mitigates LLMs’ unfaithfulness to input evidence from the perspective of cognitive inertia, providing empirical insights and robust improvements in hallucination reduction across diverse tasks and backbone models.

Based on the valuable feedback, we have made the following key additions and clarifications:

1. **Experimental Clarity and Coverage**
    - Expanded Baseline Coverage: Added standard Chain-of-Thought (CoT) baseline results and comparative analysis to enable more comprehensive faithfulness-oriented evaluation (see lines 422–425, Table 1).
    - Clarification of Backbone Models: Explicitly specify backbone LLMs and their instruct-tuning settings to enhance transparency and reproducibility (see lines 367–369).
    - Performance Analysis on HaluEval Dataset: Provided focused discussion on the source of specific performance fluctuations for the HaluEval dataset (see lines 429–464).
    - Clarification of Base vs. Prompt Instructions: Supplemented the appendix with a new instructional table showing the base and prompt instructions for summary and QA tasks across the three model bases (see Appendix C.2, Table 6).

2. **Contributions and Limitation**
    - Cognitive Inertia-Driven Hallucination Analysis: Independently analyzed the prevalence and impact of cognitive inertia-driven hallucinations among all hallucination types; introduced new error annotations and targeted metrics to highlight the contribution of LLMInertia (see Section 4.4, Table 2).
    - Comparison with Related Work: Added comparisons to two recent studies on co-occurrence and knowledge integration, clarifying the distinction of our approach and contributions (see lines 153–156).
    - Limitation Clarification: Added discussion that our method mainly addresses hallucinations from explicit co-occurrence associations due to cognitive inertia, and does not yet cover complex cases such as multi-hop reasoning or implicit evidence; noted the extensibility of the framework for future work (see lines 523–528).

3. **Robustness Verification and Methodological Detail**
    - Entity Extraction and Pipeline Validation: Provided detailed comparisons of entity extraction results between DeepSeek R1 and human annotations on representative samples, validating the reliability of this critical pipeline component (see lines 322–323, Appendix B.3).
    - Case Study Enhancements: Added detailed reasoning steps and updated figures to illustrate LLMInertia’s improvements more clearly (see Section 4.5, Appendix B.3, Figure 6).

Overall, these revisions address reviewer concerns regarding comparative baselines, experimental support, methodological clarity, and the specific impact of cognitive inertia. In summary, our work tackles the critical challenge of enhancing LLM faithfulness to explicit input evidence—a prerequisite for safe and reliable deployment in sensitive domains like healthcare and law. By systematically uncovering and alleviating the impact of "cognitive inertia," where LLMs rely too heavily on preexisting co-occurrence associations even in the presence of new or conflicting information, our study provides actionable insights into the underlying causes of hallucination and propose an adaptive, training-free approach for robust mitigation. We hope this exploration lays the groundwork for future development of more reliable, evidence-based LLMs.

We sincerely appreciate the reviewers’ thoughtful feedback and hope our comprehensive responses and revisions facilitate further discussion. We look forward to timely comments and guidance from the reviewers to drive this important direction forward.

---

### Author Response · Authors · 2025-12-01
**Rebuttal Summary for New Area Chair**

Dear Area Chairs,

We sincerely appreciate the additional time and effort you have dedicated to supporting our community during this especially challenging period. Recognizing the many demands on your time, we have prepared a concise summary of the key interactions during the rebuttal process, along with relevant excerpts from the paper and reviews, to facilitate your assessment.

Our work investigates a previously underexplored yet critical cause of unfaithfulness in large language models (LLMs): cognitive inertia. Through targeted and controlled experiments, we demonstrate that LLMs often fail to adhere to explicit input evidence due to excessive reliance on co-occurrence associations learned during pretraining. This phenomenon presents a significant barrier to safe deployment in high-stakes domains such as healthcare and law, where rigorous evidence grounding is essential.

To address this issue, we propose LLMInertia—a training-free, adaptive intervention framework that diagnoses and mitigates cognitive inertia by dynamically probing the model’s co-occurrence dependencies and injecting counter-inertial reminders into its reasoning process. Extensive evaluations across diverse datasets and backbone models consistently demonstrate substantial improvements in reducing hallucination and enhancing faithfulness, robustly surpassing existing baselines.

During the review process, multiple reviewers highlighted our novel identification and empirical study of cognitive inertia (R31P, BRzS, aoiE, MXRp), the simplicity and effectiveness of our mitigation method (BRzS, aoiE, MXRp), strong robustness across data and models (aoiE, BRzS, MXRp), and the clarity of our presentation (BRzS).

We have systematically addressed the main concerns raised by the reviewers, including clearly distinguishing our contributions from prior work, clarifying the prevalence and impact of cognitive inertia-induced hallucinations, and ensuring transparency and reproducibility. Additionally, we have expanded our discussion of current limitations in response to the valuable suggestions provided, and have considered directions for extending our approach to more complex reasoning tasks in the future, given the extensibility of our framework.

We believe that our study represents an important step toward improving the reliability of large language models across diverse, high-impact domains. We hope that our findings and our proposed adaptive, training-free framework will provide a solid foundation for future research in this area. We sincerely appreciate the area chair’s time and thoughtful consideration of our submission.

Thank you again for your patience and support, and we hope our work can contribute to the development of more reliable and faithfulness large language models.

Best regards,

Authors of Submission 8867

---

### Meta-Review · Area_Chair_UwAo · 2025-12-29

**Summary:**

The reviewers' concerns mainly focus on the limited novelty and methodological limitations of the paper. Specifically, they argue that the concept of cognitive inertia largely overlaps with existing studies on knowledge conflicts and co-occurrence biases. In addition, the proposed method is primarily validated on relatively simple entity-level questions, and its effectiveness has not been demonstrated in implicit evidence scenarios. The reviewers also point out that the experimental setup and methodological descriptions lack sufficient clarity. In particular, the dataset construction process is not adequately specified, and several critical experiments and analyses for validating hallucination errors driven by cognitive inertia are missing. Furthermore, they question whether relying solely on prompt engineering can reliably ensure that the model correctly detects and resists cognitive inertia across diverse settings.

**Reviewer Concerns:**

The rebuttal addresses several of the reviewers' concerns. The issues regarding the clarity of the proposed method and experimental details have been largely resolved. The rebuttal also provides additional clarification on the relationship between cognitive inertia and prior work on knowledge conflicts and co-occurrence biases. However, some concerns remain outstanding. The limitations of the method's applicable scenarios are not fully resolved, as the effectiveness of the proposed approach is only demonstrated under settings with explicit co-occurrence associations. Its performance in more challenging scenarios involving implicit evidence has not been validated. In addition, concerns persist regarding whether a prompt engineering–based approach alone can reliably mitigate cognitive inertia across diverse tasks and settings.

**Reviewer Scores:**

- Reviewer BRzS: The authors' responses addressed several of the reviewer's suggestions for improving the paper. As a result, the reviewer would likely maintain their original positive score.

- Reviewer R31P: The rebuttal effectively clarified the reviewer's points of confusion. Therefore, the reviewer would likely keep their original score unchanged.

- Reviewer aoiE: The major limitations raised by the reviewer have not been fully resolved. Given the reviewer's initially clearly negative score, they would likely maintain their original evaluation, with at most a very slight increase.

- Reviewer MXRp: The reviewer's concerns regarding the applicability of the proposed method were not sufficiently addressed. As a result, the reviewer’s score would likely remain unchanged.

---

### Decision · Program_Chairs · 2026-01-26

Reject